# Enhancing Antisense Oligonucleotide-Based Therapeutic Delivery with DG9, a Versatile Cell-Penetrating Peptide

**DOI:** 10.3390/cells12192395

**Published:** 2023-10-02

**Authors:** Umme Sabrina Haque, Toshifumi Yokota

**Affiliations:** 1Department of Neuroscience, Faculty of Medicine and Dentistry, University of Alberta, Edmonton, AB T6G 2H7, Canada; 2Department of Medical Genetics, Faculty of Medicine and Dentistry, University of Alberta, Edmonton, AB T6G 2H7, Canada; 3The Friends of Garrett Cumming Research & Muscular Dystrophy Canada HM Toupin Neurological Science Research, Edmonton, AB T6G 2H7, Canada

**Keywords:** antisense oligonucleotides, cell penetrating peptides, delivery, DG9 peptide, phosphorodiamidate morpholino oligomers (PMO)

## Abstract

Antisense oligonucleotide-based (ASO) therapeutics have emerged as a promising strategy for the treatment of human disorders. Charge-neutral PMOs have promising biological and pharmacological properties for antisense applications. Despite their great potential, the efficient delivery of these therapeutic agents to target cells remains a major obstacle to their widespread use. Cellular uptake of naked PMO is poor. Cell-penetrating peptides (CPPs) appear as a possibility to increase the cellular uptake and intracellular delivery of oligonucleotide-based drugs. Among these, the DG9 peptide has been identified as a versatile CPP with remarkable potential for enhancing the delivery of ASO-based therapeutics due to its unique structural features. Notably, in the context of phosphorodiamidate morpholino oligomers (PMOs), DG9 has shown promise in enhancing delivery while maintaining a favorable toxicity profile. A few studies have highlighted the potential of DG9-conjugated PMOs in DMD (Duchenne Muscular Dystrophy) and SMA (Spinal Muscular Atrophy), displaying significant exon skipping/inclusion and functional improvements in animal models. The article provides an overview of a detailed understanding of the challenges that ASOs face prior to reaching their targets and continued advances in methods to improve their delivery to target sites and cellular uptake, focusing on DG9, which aims to harness ASOs’ full potential in precision medicine.

## 1. Introduction

The advancement of antisense oligonucleotides (ASOs) has brought about a profound change in the field of genetic therapeutics, offering a promising avenue for addressing a diverse array of diseases on a molecular level. ASOs are short synthetic nucleic acid analogs that offer a revolutionary means to modulate gene expression by precisely interacting with RNA transcripts. The history of ASO can be traced back to the pioneering work of Zamecnik and Stephenson in early 1970, who first proposed the concept of using synthetic oligonucleotides to regulate eukaryotic gene expression in cultured cells through sequence-specific hybridization with RNA [1,2]. Later, the pharmacokinetic properties of ASOs, such as stability, reduced susceptibility to nuclease degradation, specificity, and cellular absorption, have been greatly improved by developments in oligonucleotide chemistry, including the introduction of chemical modifications and different backbone structures, which transformed them from theoretical concepts into potentially effective therapeutic agents [3]. 

ASOs have been successfully employed in treating a wide range of diseases, including Duchenne Muscular Dystrophy (DMD), spinal muscular atrophy (SMA), amyotrophic lateral sclerosis (ALS), and many more. This success led to the regulatory approval of 10 ASO-based drugs [4] and many antisense drug candidates for clinical trials to treat cardiovascular, metabolic, endocrine, neurological, neuromuscular, inflammatory, and infectious diseases [5]. This demonstrates the dynamic nature of ASO-mediated therapy. Despite being a promising approach, it is widely accepted that the delivery of ASO treatments to specific tissues is limited by factors such as intracellular trafficking, degradation in biological fluids, and transportation across cellular barriers [6]. Although chemical modifications have significantly improved their metabolic stability as well as their affinities for RNA targets and have, to some extent, reduced off-target effects, no chemical modification has significantly improved cellular uptake or tissue targeting.

Cell-penetrating peptides (CPPs) or peptide transduction domains (PTDs) are one of the many approaches that have been developed to improve the delivery of oligonucleotides. CPPs are small peptides with the ability to transport cargos, including ASOs, across cellular barriers and hereby offer the potential to improve ASOs’ cellular uptake and intracellular distribution, enhancing therapeutic outcomes and reducing the required dosage [7]. The initial CPP was introduced several decades ago, and ever since, there has been an ongoing endeavor to enhance cell-penetrating peptides for improved oligonucleotide delivery and enhanced pharmacological properties [8].

Particularly in the context of phosphorodiamidate morpholino oligomers (PMOs), R6G, PiP (PNA/PMO Internalizing Peptides), and DG9 have captured interest among the CPPs for their potential to improve ASO-mediated therapy. PMOs have shown effectiveness in treating genetic diseases, but their poor cellular absorption continues to be a major drawback. Due to its high efficacy and low toxicity, DG9 has become a promising CPP for improving the intracellular transport of PMOs since it holds the prospect of improved therapeutic advantages [9,10]. This review offers a thorough analysis of ASO therapies and their difficulties, highlighting the potential contribution of CPPs, particularly DG9, to overcoming these difficulties and improving ASO efficacy. Through an exploration of CPP-mediated ASO delivery intricacies and focusing on the remarkable properties of DG9, this review seeks to highlight the potential of this approach to transform ASO-mediated therapy more effectively.

## 2. Why Antisense Technology?

With their ability to precisely target disease-causing genes at the RNA level, antisense oligonucleotides (ASOs) have become an important tool in the development of therapeutics. ASOs show promise in the treatment of a wide range of illnesses, such as cancer, viral infections, genetic disorders, and neurological problems. Currently, 15 oligonucleotide therapeutics have received approval from the Food and Drug Administration (FDA, USA), the European Medicines Agency (EMA), and/or the Japanese Ministry of Health, Labour, and Welfare, and most of them have received approval in the past 4 years [11] (Table 1). Compared to small molecules antisense technology’s unprecedented specificity, ability to modulate gene expression, variety of target types, potential for personalized therapy, disease modification abilities, and documented clinical effectiveness make antisense technology an appealing strategy for therapeutic research.

## 3. The Mode of Action of Antisense Oligonucleotide

Antisense oligonucleotides (ASOs) are synthetic, single-stranded nucleic acid molecules targeted for mRNA, generally comprised of ~18–30 nucleotides with a variety of chemical structures [12]. ASOs form a DNA–RNA hybrid by binding specific RNA sequences through Watson–Crick base pairing to modulate gene expression [13]. The functional mechanism of ASO can be broadly categorized into two main modes of action: RNase H-mediated degradation and steric hindrance [13].

RNase H-mediated degradation: When DNA-based oligonucleotides, also known as gapmers, bind to their respective mRNA sequences, they can recruit endogenous RNase H enzymes. RNase H recognizes the RNA–DNA duplex and catalyzes the degradation of RNA, leading to the reduction in the target RNA and gene silencing (Figure 1a) [14]. This strategy has been employed widely to suppress disease-causing or disease-modifying genes. Fomivirsen, mipomersen, and inotersen are the three RNase H-competent ASOs that have so far acquired regulatory approval [14].

Steric hindrance: Apart from RNase H-mediated breakdown, ASOs can interfere with RNA–RNA or RNA–Protein interaction by blocking certain regions within the target transcript. This results in the prevention of translation rather than the lowering of transcript levels [15]. The best-known application of this mode of action is splicing modulation, which can cause either exclusion (exon skipping) or retention (exon inclusion) of specific exon/exons by targeting splice sites or exonic/intronic inclusion signals, respectively [16,17]. Typically, this approach can be used both for restoration of the translational reading frame to have functional protein synthesis or for disruption of translation of the target gene [18,19] (Figure 1b). Eteplirsen, golodirsen, nusinersen, viltolarsen, casimersen, milasen, and atipeksen are the splice-switching ASOs that have received FDA approval to date [12,20,21,22].

## 4. Molecular Mechanism of Cellular Uptake and Intracellular Distribution of Antisense Oligonucleotides

The effectiveness of antisense oligonucleotides (ASOs) as therapeutic agents depends significantly on cellular uptake and intracellular distribution. To have the desired effects, ASOs must efficiently penetrate cells and locate their target locations. After intravenous, subcutaneous, or direct administration, ASOs reach the bloodstream, where they can be broken down by nucleases [23]. Once they reach the target organ, the cellular uptake process can be achieved in several ways, such as phagocytosis, macropinocytosis, micropinocytosis via clathrin and caveolin-independent pathways, caveolar internalization, and classical clathrin-mediated endocytosis. Following cellular uptake, ASOs are internalized into early endosomes and then late endosomes, regulated by Rab, SNARE, and tethering proteins. A percentage of ASO drugs, possibly a very tiny portion, are released from late endosomes into the cytoplasm, where they target mRNAs or pre-mRNAs in the cytoplasm or the nucleus to carry out their therapeutic effects. Nuclear entry can be actively mediated by the nuclear pore mechanism or passively via simple diffusion [24]. Many small cellular proteins, such as COPII, can facilitate nuclear trafficking. However, the process is not entirely known [23]. The target of different ASOs is located at different subcellular sites. For RNase H-mediated mRNA degradation, the ASO drugs need to reach either the cytoplasm (ribosomes) or the nucleus [25]. In contrast, for exon skipping/inclusion, ASOs must be present in the spliceosomes of the nucleus [26]. Another percentage of ASO medications enter lysosomes or are subsequently expelled from the cell by one of three hypothesized mechanisms: membrane leakage, back-fusion-mediated release, or vesicle-mediated release [23]. It is essential for ASOs to avoid or circumvent lysosomal degradation to maintain their integrity and efficacy. Apart from that, cellular uptake and intracellular localization of ASOs can significantly vary based on the ASO chemistry, cell type, and specific cellular conditions.

## 5. Challenges Associated with ASO Delivery

Although ASOs have great potential as therapeutic agents, their efficient delivery faces several difficulties. These difficulties are associated with the physiochemical characteristics of ASO molecules, such as their large size, molecular weight (single-stranded ASOs are ~4–10 kDa, double-stranded siRNAs are ~14 kDa), and negative charge, which hinders passive diffusion across the cell membrane. ASOs predominantly rely on endocytosis for cellular uptake, which might be ineffective and lead to entrapment in endosomes or lysosomes, leading to lysosomal degradation. So, once inside the cell, ASO must escape endosomal entrapment to gain access to the target region in the cytoplasm or nucleus [27]. Apart from that, for the systemically administered ASOs to be effective, they need to avoid renal clearance [28,29], resist nuclease degradation both in the extracellular fluid and intracellular compartment [30], and avoid removal by the reticuloendothelial system, which includes mononuclear phagocytes, liver sinusoidal endothelial cells, and Kupffer cells [31]. A study reported that intravenous administration of an AON resulted in 40% and 18% accumulation in the liver and kidneys, respectively [32]. Recently, ASOs have also been developed for the treatment of central nervous system (CNS)-related diseases. The additional barrier—in this case, ASOs—has to cross the blood–brain barrier (BBB) or brain–cerebrospinal barrier before they can distribute within the CNS. The vascular barriers of the nervous system are composed of a monolayer of endothelial cells forming tight junctions through interactions of cell adhesion molecules, which prevents most of the ASOs from reaching the CNS after systemic injection [33] (Figure 2).

Due to these challenges, to date, most of the approved oligonucleotide treatments are delivered either locally (for example, to the eye or spinal cord) or to the liver. The eye is chosen as a target for ASO delivery (for example, Pegaptanib and Fomivirsen) due to its accessibility, anatomical considerations, and immune-privileged status [12]. Although ocular delivery of ASOs has benefits, there are still obstacles to be overcome, including getting through anatomical obstacles (such as the blood–retinal barrier), maximizing ASO stability, and pharmacokinetics for long-lasting therapeutic effects. For ASOs targeting the CNS, direct delivery into the cerebrospinal fluid via lumber puncture is most commonly used (for example, Nusinersen) [34]. However, it should be noted that this method requires expertise and specialized equipment and carries a small risk of complications associated with invasive procedures.

## 6. Strategies to Enhance the Stability and Delivery of Antisense Oligonucleotides

### 6.1. Chemical Modification

Antisense oligonucleotides were initially employed as synthesized, unaltered DNA, which turned out to be extremely vulnerable to exonuclease and endonuclease degradation [35] (Figure 3). Chemical modifications of antisense oligonucleotides can enhance stability, improve target binding affinity and biodistribution, and provide protection against nuclease-mediated degradation. Modification of the nucleic acid backbone, the ribose sugar moiety, and the nucleobase itself have been extensively employed to improve the drug-like properties of antisense oligonucleotides [29,36].

#### 6.1.1. Backbone Modification

Backbone modifications involve changing the repeating sugar-phosphate units that make up the phosphodiester backbone of ASOs. Typical changes to the backbone include the incorporation of phosphorothioate (PS) linkages, in which one of the non-bridging oxygen atoms of the inter-nucleotide phosphate group is replaced with sulfur [37] (Figure 3c). Phosphorothioate (PS) belongs to the first generation of ASOs that work by an RNase H-mediated mRNA cleavage-based mechanism and do not disrupt RNase H activity [38]. Unmodified ASOs are reported to be degraded within 30 min in serum [1], whereas oligonucleotides with these modifications are more stable, with reported half-lives of 9 h in human serum [39]. Scavenger receptors (such as the stabilins STAB1 and STAB2) also take up sulfated molecules, such as oligonucleotides with PS linkages, and facilitate their internalization into organs like the liver [32,40,41]. Apart from that, the incorporation of PS linkages increases the binding of ASO to proteins in both plasma and within cells, which significantly improves drug pharmacokinetics by reducing renal clearance and increasing circulation time [42]. However, high PS concentrations have been shown to cause cytotoxic effects, which are assumed to be due to protein binding [43,44,45]. In addition to that, PS backbone modifications have been found to reduce the binding affinity of the oligonucleotide for its target [12].

#### 6.1.2. Ribose Sugar Modification

Oligonucleotides are frequently modified at the 2′ position of the ribose sugar to provide resistance to enzymatic degradation, improve stability in plasma, and increase tissue half-lives. 2′-O-methyl (2′-OMe), 2′-O-methoxyethyl (2′-MOE), and 2′-Fluoro (2′-F) are among the most used 2′ substituents (Figure 3b) [46]. 2′-ribose-modified ASOs are used to sterically block oligonucleotides or flanking sequences in gapmer ASOs, as these modifications are not compatible with RNase H activity [12].

In comparison to unmodified phosphorothioates, 2′-O-methyl (2OMe) and 2′-O-methoxy-ethyl (MOE) have been shown to increase hybridization affinity to their target RNA and decrease sequence-independent toxicity arising from the PS backbone [47,48,49,50]. Due to their extensive success and effectiveness, among the fifteen FDA-approved drugs, five use 2OMe or 2′MOE chemistry (for example, Pegaptanib, Mipomersen, Nusinersen, Patisiran, and Inotersen) [12].

Locked nucleic acids (LNA) are a type of 2′-modification in which the 4′-carbon is linked to the 2′-hydroxyl group and have also been utilized in steric block ASOs, such as miRNA inhibitors (Figure 3b). LNAs offer increased resistance to nucleases [51] and exhibit significantly improved hybridization compared to other 2′-modifications [52,53]. However, they are associated with more severe toxicological issues in systemic treatment [54]. Furthermore, a loss in target specificity can also be due to the strong affinity of LNAs [53,55].

Other sugar modifications that are less frequently used are tricyclo-DNA (tc-DNA) and S-constrained-ethyl (cEt) (Figure 3b). For tc-DNA modification, this adds an ethylene bridge fused with a cyclopropane unit, which results in a more stable duplex formation [56]. When tested in cells, Tc-DNA was found to be more effective at correcting splicing than a 2OMe-PS oligonucleotide and to be stable in serum [57]. However, tc-DNA has only been used in a very small number of studies up to this point. Finally, the cEt-modified antisense oligonucleotides show a similar binding affinity to LNA but a better toxicity profile [58] and have recently shown good promise in a humanized mouse model for HD (Huntington disease) [59].

#### 6.1.3. Nucleobase Modification

ASOs’ characteristics can also be enhanced by the introduction of nucleobase modifications in ASO to achieve optimized Watson–Crick base-pairing and thereby control the melting temperature of the ASOs [36] (Figure 3d). Alteration of the nucleobase chemistry results in a thermally more stable ASO-target duplex by increasing the affinity towards the target. The thermal stability of splice-switching ASOs plays a vital role, as a stronger hybridization between ASOs and their target can hinder the formation as it directly impacts their ability to effectively block the splice site or hinder the assembly of the ribosomal complex, ultimately preventing translation and executing the therapeutic effects [60]. Among nucleobase modifications, the incorporation of cytosine analogs has been extensively used. 5-methylcytidine and 5-methyluridine/ribothymidine) has the effect of increasing the oligonucleotide melting temperature by ~0.5 °C per substitution [36] (Figure 3d). Apart from that, 5′-methyl cytosine-based analogs were used to reduce the immunological stimulation caused by CpG dinucleotide-mediated toll-like receptor activation [61,62].

#### 6.1.4. Alternative Chemistries of ASOs

Apart from the modifications mentioned above, other chemistries have also been explored for the improvement of ASOs’ drug-like properties, such as PNA (Peptide Nucleic Acid) and PMO (Phosphorodiamidate Morpholino Oligomer) (Figure 3e). PNA is a synthetic nucleic acid analog in which the sugar-phosphate backbone is replaced with a peptide-like backbone, which makes them uncharged [63]. As a result, PNAs have a high binding affinity and are resistant to enzymatic degradation [64,65]. These are mostly implemented in splicing modulation approaches or translation inhibition, as they are unable to activate the RNase H enzyme. A clear shortcoming of this type of modification is its poor cellular uptake and water insolubility [66,67]. PNAs are found to be rapidly cleared when administered peripherally [68], and these poor pharmacokinetic properties are the main reason for their limited in vivo use thus far.

Another strategy that has been explored is the use of PMO synthetic backbone modification, in which the five-membered ribose heterocycle is replaced by a six-membered morpholine ring structure with phosphorodiamidate linkages [69]. Similar to PNA, PMOs are neutrally charged and work by steric hindrance or splice modulation to provide an antisense effect [70]. The absence of a carbonyl group provides PMO resistance against protease and nuclease degradation [70], and their neutral charge makes them less susceptible to activating immune responses [71]. Apart from that, studies have shown that the administration of multiple high doses can be achieved with minimal toxicity [72].

To date, four PMO drugs have been approved by the FDA: eteplirsen, golodirsen, viltolarsen, and casimersen (Eteplirsen targets exon 51, Golodirsen and Viltolarsen target exon 53, and Casimersen targets exon 45 of the dystrophin mRNA) for Duchenne Muscular Dystrophy [73,74] (Table 1). The first in vivo exon skipping of *Dmd* using 2′-OMePS ASO chemistry was documented in a work by Lu et al. [75]. In another work, exon 23 skipping was successfully induced, and dystrophin production was restored in *mdx* mice by the intramuscular administration of leashed PMOs (PMOs annealed to complementary anionic oligonucleotides to increase delivery). Even two weeks after the injection, the skipped transcript was observed, which was not the case with 2′-OMePS ASOs [76]. An additional investigation on CXMDJ dogs discovered that intramuscular or intravenous delivery of a 3-PMO cocktail promotes in-frame 6 to 8 exon skipping with 61–83% exon skipping efficiency two weeks after intramuscular injection, which results in approximately 25–50% dystrophin protein restoration [77].

In addition to the DMD, the neutrally charged PMO chemistry was also being studied by several groups for the treatment of central nervous system (CNS) disease. For Spinal Muscular Atrophy, a neurodegenerative disease, peripheral administration of Nusinersen, an MOE-based drug, has already been proven to rescue the SMA phenotype [78,79] (Table 1). However, with peripheral delivery or intrathecal injection being more invasive, the focus is now to modify the ASO in such a way that systemic injection can reach the CNS by crossing the blood–brain barrier (BBB). There is a length-dependent effect of PMO ASO. Two separate studies showed that a single intravenous injection of longer PMO (25 mer) in a severe SMA mouse model can increase the survival rate. Based on the experimental results, they emphasized the superiority of the morpholino ASOs because of their lower toxicity, increased SMN levels, and prolonged survival [80,81]. In addition to that, another study proved PMO can more readily cross the immature BBB into the CNS [72].

However, the main pharmacokinetic shortcomings associated with PMO are low efficacy, which is related to its rapid clearance from the bloodstream, poor uptake in tissues like the skeletal muscle, and endosomal entrapment [82,83]. As PMOs are uncharged nucleic acid molecules, this provides the opportunity to covalently conjugate them to charged delivery-promoting moieties such as cell-penetrating peptides (CPPs) for enhanced delivery, as mentioned below [69,84].

### 6.2. Bioconjugates

While chemical modifications are required to protect ASOs from exonucleases and prolong their stability, the next challenge is ASO passage across biological barriers. These barriers include the vascular endothelial barrier, cell membranes, and intracellular compartments. Additionally, achieving specific cell/tissue targeting and a reduction in clearance from circulation is essential [85]. Improving ASO delivery potential can be achieved through the conjugation of different moieties that can direct the drug to specific tissues and enhance internalization. Bioconjugates are distinct molecular entities with precise stoichiometry, which ensures well-defined pharmacokinetic properties and simplifies large-scale synthesis. Additionally, bioconjugates tend to have a small size, which often results in favorable biodistribution profiles [12]. Bioconjugates usually promote interaction with cell-type-associated receptors, consequently enhancing delivery to the target tissue and internalization by receptor-mediated endocytosis [86]. There are different types of conjugates available, including lipid-based bioconjugates (e.g., cholesterol and its derivatives) [87,88,89], peptide-based bioconjugates (e.g., cell-penetrating peptides) [90,91,92,93,94,95], aptamers [96], antibodies [97,98], sugars (for example, N-acetylgalactosamine (GalNAc)) [99,100], and polymers (e.g., PEG) (Table 2). The selection of the appropriate bioconjugate depends on several factors, including the application goals, specific requirements of the ASO delivery system, the intended therapeutic application, and safety considerations. Due to the effectiveness of bioconjugates in increasing the efficacy of ASO delivery, bioconjugated compounds are present in four of the five FDA-approved siRNA medications [101].

#### 6.2.1. Cell Penetrating Peptides

Cell-penetrating peptides (CPPs, also known as protein transduction domains) are short (fewer than 30 amino acids) cationic, amphipathic, or hydrophobic peptides that translocate small drugs/cargo across cell membranes and biological barriers [101,102]. CPPs were not just recently discovered; they were first identified in 1988 when two research groups identified that the transactivator of transcription (TAT) protein of HIV can cross the cell membranes and can be efficiently internalized by cells due to the strong electrostatic interactions with heparin sulfate during endocytosis [103]. Later, another group of scientists found thirteen amino acid sequences that align with residues 48–60 of TAT and play a key role in cellular uptake. Subsequently, peptides exhibiting cell membrane translocation like TAT were commonly categorized as cell-penetrating peptides (CPPs) [104].

CPPs can be classified into cationic CPPs, amphipathic CPPs, and hydrophobic CPPs based on their physicochemical characteristics [105]. Cationic CPPs are mainly composed of basic amino acids such as arginine and lysine [104]. Poly-arginine stretches exhibit the utmost capacity for cellular uptake and hold considerable therapeutic potential [105]. However, higher values of arginine are related to irreversible side effects [106]. Several studies have shown that positively charged CPPs interact with negatively charged carboxylic, phosphate, and sulfate groups of the cell membrane and eventually mediate internalization by endocytic pathways [104,107]. Some examples of cationic CPPs include TAT, penetratin, and polyarginine [104]. Amphipathic CPPs contain polar and non-polar amino acid regions. Generally, the non-polar region is made up of valine, leucine, and alanine, whereas the polar region is made up of lysine and arginine. Amphipathic peptides have been found to play a role both in cellular internalization and endosomal escape [107]. This group of CPPs makes up more than 40% of all the CPPs that have so far been identified [105]. Some of the examples of amphipathic CPPs are MAP, transportan, and Pep-1 [103]. Compared to other types of CPPs, there are relatively few hydrophobic CPPs that are typically composed of a large number of non-polar residues or only a few charged amino acids (less than 20% of the sequence) [105]. Hydrophobic CPPs usually interact with the hydrophobic region of the cellular membrane and probably translocate by an energy-independent mechanism. Examples of such natural hydrophobic CPPs include K-FGF, C105Y, and gH625 [104]. However, the peptide sequence of hydrophobic CPPs has not been found to significantly affect cell uptake [108]. CPPs can function either via receptor-mediated endocytosis, where CPPs trigger endocytosis by directly binding to the specific cell surface receptor, or via direct translocation, where CPPs cross the cell by creating transient pores on the cell membrane and deliver the cargo inside the cytoplasm or nucleus [101].

## 7. Overcoming the Limitations of PMO by Conjugating It with Cell Penetrating Peptides

As mentioned earlier, PMOs show low efficacy as therapeutic agents due to their poor cellular uptake, less permeability of membrane barriers, rapid clearance from the systemic circulation, inability to cross blood–brain barriers, and the requirement of repetitive administration and/or a high dosage of the drug for executing its function. Apart from that, due to the hydrophobicity of the plasma membrane and the neutral charge in PMO, only small portions of internalized PMOs can escape endosomes and reach their intended target [83]. A promising utilization of CPP is their ability to directly conjugate with neutrally charged PMO and PNA and increase the delivery efficacy [109,110,111].

A promising utilization of CPP is their ability to directly conjugate with neutrally charged PMO and PNA using several methods, including maleimide linkage, disulfide linkage, click chemistry, or amide linkage. It enhances the pharmacokinetic properties of PMO and PNA. Among various therapeutic purposes, this approach has been extensively explored, mostly for Duchenne Muscular Dystrophy (DMD). This affects approximately 1 in 3500 newborn boys and is caused by out-of-frame deletions in the *Dmd* gene, resulting in the loss of dystrophin, the structural muscle protein [112]. Lack of dystrophin results in progressive muscular degeneration, which impairs ambulation and causes mortality from cardiac and respiratory failure [113]. The mRNA reading frame around the deletion can be restored by the “exon-skipping” approach, where pre-mRNA splicing is modulated to produce smaller but functional proteins. This approach has been used successfully with naked PMO and resulted in the conditional approval of four PMO-based drugs for DMD (e.g., eteplirsen, golodirsen, viltolarsen, and casamirsen) [73,74]. Eteplirsen has been found to restore an average of 0.9% dystrophin to normal levels after 180 weeks of treatment, which indicates low treatment efficacy despite the safe profile of this drug [102]. For golodirsen, according to the trial results, dystrophin expression increased by ~0.9% after the demonstration of the drug [73]. The casimersen-treated group saw a 0.81% increase in dystrophin production [114]. Whether such a tiny increase in dystrophin expression is enough to slow down disease progression and provide clinical benefits is still a big question. Apart from that, viltolarsen has limited efficacy in cardiac tissue due to poor uptake. As the primary cause of mortality in the DMD patient population is cardiorespiratory complications, the low efficacy of the drug in the heart is a serious concern in exon-skipping therapy [115]. Therefore, there is still a need for a more potent substance to raise dystrophin levels and thereby maximize the functional advantages of this strategy.

Conjugation of CPPs with PMO is one such approach to improving PMO delivery. This strategy was first demonstrated with an arginine-rich peptide (RXR)4, which was administered to the *mdx* mouse model of DMD in a variety of doses, time intervals, and delivery methods. It was observed that a single intravenous administration can cause high dystrophin exon skipping in skeletal muscle, the diaphragm, and for the first time in the heart [116]. Another arginine-rich peptide, (RXRRBR)2 peptide (B-peptide), was identified from a screen using the EGFP-654 splicing reporter mouse model to ensure PPMO entry to cells and notable exon-skipping in the heart after retro-orbital injection, resulting in improved cardiac function, specifically end-systolic volume and end-diastolic volume and resistance to dobutamine [117]. In another study, intravenous injection of a single 25 mg/kg dose of B-peptide conjugated PMO into *mdx* mice confirmed approximately 50% of wild-type dystrophin levels along with restoration in cardiac function [116,117] and improved muscle function. In contrast, weekly administration of naked PMO at 200 mg/kg for 12 weeks could only achieve 10% wild-type dystrophin levels [118]. Fusion of muscle-specific peptide (MSP) with B-peptide through a phage display has been found to improve activity 2- to 4-fold after multiple 6 mg/kg doses [119]. Interestingly, another study revealed that a specific orientation (B-MSP-PMO) can lead to a 2–5-fold improvement in skeletal muscle restoration compared to B-PMO [120]. Additionally, B-PMO has also been used for research in canine models of DMD that better mimic the pathophysiology of human illness and serve as a more rigorous evaluation of the efficacy of CPP-PMOs in restoring dystrophin expression. A repeat low-dose (4 mg/kg per ASO) B-PMO intravenous injection has been found to restore 5% dystrophin of wild-type levels throughout the body, including in the heart, where improvement of cardiac conduction defects was seen after therapy [121]. Another arginine-rich peptide, R6G, is also currently being explored for the treatment of DMD [122]. R6G peptide is a modification of the conventional R6 peptide with the glycine residue that has been extensively studied for various neuromuscular disorders [123,124]. When conjugated with PMO, it has shown promise for exon-skipping efficacy, specifically in cardiac muscle [122].

Recent research has led to the development of several peptide series known as “Pip’s” (PMO/PNA internalization peptides), which are generated from the parent peptide penetratin [125,126] and consist of the amino acids arginine (R), 6-aminohexanoic acid (X), and ß-alanine spacer (B), with an internal core containing hydrophobic residues [12]. The most recent Pip-PMO conjugates are significantly more effective than naked PMO and, more critically, reach cardiac muscle following systemic administration in dystrophic animal models. A single intravenous injection of the Pip5e peptide-conjugated PMO induced the highest amounts of exon skipping and dystrophin restoration throughout the body, including in the hearts of *mdx* mice [127]. To increase homogenous dystrophin repair and to target the heart muscle more effectively, the Pip6 series of peptides were generated by further iterations of the core design [128]. In a study, it was observed that inversion of the Pip5e-PMO hydrophobic core (Pip6a) resulted in a cardiac dystrophin recovery score of up to 37% in the *mdx* animal model [92]. In another study by the same group, it was demonstrated that administering Pip6f-PMO (scrambled peptide core) can increase the levels of the protein dystrophin by up to 28% in the hearts of *mdx* mice who had previously undergone a forced exercise regimen to cause changes resembling the DMD cardiac phenotype [129]. Additionally, injection of Pip2a or Pip2b conjugated PPMOs in the tibialis anterior of the *mdx* mouse has also been found to induce an effective exon 23 skipping and a noticeable increase in dystrophin rescue [130]. Another CPP created to target muscle is M12, which was discovered through a phage display conducted on C2C12 myoblasts upon conjugation to PMO. M12 achieved approximately 10–25% of wild-type dystrophin levels following a single systemic administration, although at dosing levels 5- to 6-fold higher than those required for comparable efficacy [102].

CPP-PMO has also been used as a therapeutic approach for myotonic dystrophy type I, where a CTG expansion in the DMPK gene’s 3′ untranslated region causes a pathogenic transcript that interacts with RNA-binding proteins like muscleblind-like 1 (MBNL1) to cause widespread aberrant splicing abnormalities. Systemic administration of B-PMO targeting this repeat element causes the blocking of Mbnl1 sequestration, resulting in normal nuclear distribution and subsequent correction of abnormal RNA splicing, including for the chloride channel 1 gene, which is a primary contributor to myotonia [131].

One of the biggest challenges of nucleic acid therapy is crossing the blood–brain barrier to reach the central nervous system (CNS) after systemic delivery. CPPs have been identified as promising medicines in the treatment of central nervous system (CNS) diseases due to their demonstrated transmembrane transporting ability. It is assumed that small-size cationic or amphipathic CPPs may exhibit greater affinity for negatively charged endothelial cells on the blood–brain barrier [132,133]. CPP-PMOs have recently been investigated in preclinical models of spinal muscular atrophy (SMA), an autosomal recessive neuromuscular disorder that results in premature death [134]. This disease is caused by mutations in the survival of the motor neuron 1 (SMN1) gene. A paralogous gene, SMN2, encodes a vital SMN protein but generates only minimal levels due to a sequence variant leading to the exclusion of exon 7 from approximately 90% of mature transcripts. Consequently, a truncated, non-functional protein is produced [135,136]. To address the functional deficiency caused by the loss of SMN1 protein in patients, ASOs have been employed to facilitate the inclusion of exon 7 in SMN2 transcripts, thereby enhancing the production of SMN2 protein [137]. However, the limited delivery of the currently used ASO in the rostral spinal cord and brain has reduced therapeutic efficacy. Nusinersen, a modified 2′-MOE PS ASO, has been recently approved by the FDA for the treatment of SMA. Intrathecal injection of Nusinersen can significantly improve motor function and increase the lifetime of SMA patients [138,139]. However, this procedure is invasive and is linked to unpleasant post-lumbar puncture adverse effects for the patients [137]. Therefore, to address this, PPMO trials have been conducted. Intravenous administration of Pip6A-PMO in the Taiwanese severe SMA mouse model increased mean survival and SMN2 expression in the brain and spinal cord and improved neuromuscular junction morphology [140]. Due to the mouse model’s severity, the drug has to be administered before postnatal day 2 to demonstrate functional benefit. It is likely that the BBB may not be fully formed at that time and, as a result, does not accurately represent the clinical condition for therapeutic intervention [102]. To prove the blood–brain barrier crossing capacity of PPMO, a study has been conducted where symptomatic SMA mice were administered RXR-MO and r6-MO (morpholino oligomer) conjugates intraperitoneally at PD-5 with a completely closed BBB. The treated mice showed improved median survivals of 41.4 and 23 days, respectively, which is significantly higher compared to the naked MO (~17 days). Additionally, RXR-MO and r6-MO conjugates were found in the central nervous system in a symptomatic phase. Pathological studies demonstrated that CPP-MOs mitigate the degradation of neuromuscular connections more efficiently than scrambled or naked MOs [124]. Another study demonstrated that a derivative of an ApoE could induce a 0.25-fold increase in exon 7 inclusion in the pre-mRNA of the spinal cord and, to a lesser extent, in the brain of a spinal muscular atrophy mouse model, improving the diseased mice’s phenotype [141].

CPP-PMO strategies have also been developed for the treatment of other neurological diseases like Huntington’s disease (HD) and Amyotrophic Lateral Sclerosis (ALS) [83], as well as for use as antibacterial agents because ASOs by themselves are not very effective at penetrating bacterial cell walls [102]. It is evident that CPPs have a great deal of therapeutic potential in delivering and increasing the efficacy of ASOs specifically the PMO-based strategy.

## 8. DG9: A CPP for Enhancing the Delivery and Cellular Uptake of ASO and Proteins

Although CPPs hold promise in facilitating the transport of biologically active cargo across cell membranes, including the notorious blood–brain barrier and other challenging barriers within the body, they also pose a number of difficulties and issues that require careful study. The primary obstacle to completing clinical trials for PPMO-based medications right now is their toxicity and immunogenicity. Toxicity can be variable depending on several factors, including species, treatment duration, frequency of systemic administration, dosage, exons skipped, and the cationic nature of the peptide [83]. Additionally, first-generation arginine-rich peptides were found to be more immunogenic than PMOs [142], suggesting that the toxicity may result from immunogenic processes such as complement activation [121,143]. Due to severe side effects, a preclinical experiment using an arginine-rich PPMO by Sarepta had to be stopped. It is assumed that the side effects were partially attributable to the high dosage employed [144]. In a separate study, rats given high doses of B-peptide-PMO experienced a loss of body weight and an increase in serum blood urea nitrogen and creatinine in a dose-dependent manner, indicating decreased renal output [145]. Therefore, the quest for cell-penetrating peptides is still ongoing in order to overcome the challenges. A peptide found recently in this search is DG9.

DG9 is a cell-penetrating peptide derived from the protein transduction domain (PTD) of the human Hph-1 transcription factor, which facilitates the cell membrane penetration of its protein cargos in the lungs (Figure 4). Two of these Hph-1 domains constitute DG9 [9]. In a study by Choi et al., it was shown that intraperitoneal injection of fusion proteins conjugated with the Hph-1 domain has enhanced delivery in a wide range of organs, including the heart and brain, which are apparently challenging to deliver. Additionally, according to the study, cell viability was not affected, and behavioral abnormalities, cytotoxic effects, and immunogenicity were not observed after 1.6 mg/kg of intravenous administration of Hph-1-fused protein into mice for 14 days or 100 μg of intraperitoneal injection two times a week for two weeks [146]. The same author later reported another study where they used the same protein transduction domain (PTD) of the human Hph-1 transcription factor, but this time with two tandem sequences (HHph-1-PTD) and fused it with Foxp3 (the target protein of the study) protein to increase the cell permeability of Foxp3. In an in vitro study, HHph-1-Foxp3 was detected in the nucleus as well as in the cytoplasm within 30 min of transduction, suggesting that Foxp3 protein is efficiently delivered to cells and is localized in the nucleus. The delivery efficacy of HHph-1 was also proved in vivo, as HHph-1-Foxp3-treated mice lived longer and their phenotype improved compared to the control groups. They also found that two repeats of Hph-1-PTD (HHph-1) resulted in optimal intracellular transduction and rapid delivery compared to one Hph1 domain [147].

A separate study reported that a conjugate of PMO and a unique peptide, derived from a human T cell and a near dimer of the PTD, are at least 10- to 100-fold more efficient than the prior peptides at delivering the PMO into bacteria and ultimately causing bacterial death. The only difference between the DG9 and the peptide used in the study is that only L forms of amino acid residues were used in the peptide [148]. Kim et al. previously demonstrated that DG9 can deliver a PMO to the zebrafish heart and can cause strong exon skipping in the heart while also inducing exon skipping at significantly greater levels in the skeletal muscle [149].

The FDA has approved exon skipping as a promising therapy for DMD, which utilizes phosphorodiamidate morpholino oligomer (PMO) to target and modulate gene expression. Yokota’s research group has identified DG9 peptide conjugation as a powerful way of enhancing the exon-skipping efficacy of PMO in vivo [9]. The positive aspect of the DG9 peptide used in this study is that it has a potentially better toxicity profile compared to other peptides. As mentioned earlier, peptide-conjugated PMOs have been found to induce dose-dependent toxic effects in preclinical studies, which are thought to be linked with their amino acid compositions [83,144,150]. It has also been reported that substituting D-amino acid for L-amino acid in polymer-peptide conjugates attenuates anti-polymer antibody generation and toxicity and exhibits good tolerance in vivo even after repeated administration [151]. Therefore, certain L-arginine residues in DG9 were converted to D-arginine (DG9 (sequence N-YArVRRrGPRGYArVRRrGPRr-C; uppercase: L-amino acids, lowercase: D-amino acids)) [9]. This conversion has been shown to improve the viability of peptide-conjugated PMO-treated cells in vitro, along with increasing serum stability [152]. Additionally, this DG9 does not contain any 6-aminohexanoic acid residues (often represented by “X” in peptide sequences), which have also been linked to higher toxicity [152]. In the study of DG9-PMO-mediated efficient exon skipping by Lim et al., it has been demonstrated that retro-orbital injection of DG9-conjugated PMO into hDMDdel52; *mdx* mice can increase skipping efficiency by a factor of 2.2 to 12.3-fold and 14.4-fold compared to the unconjugated PMO. This resulted in a dystrophin restoration amount of 3% and 2.5% of wild-type levels in skeletal muscles and the heart, respectively. Skeletal muscles produced 2.8 to 3.9% more dystrophin and had an exon 51 skipping level of 55 to 71% after receiving repeated injections of DG9-PMO once each week for three weeks. Most notably, hDMDdel52; *mdx* mice treated repeatedly with DG9-PMO showed a considerable improvement in forelimb and total limb grip strength, indicating the improvement of the muscle function of the treated mice. Additionally, the tibialis anterior DG9-PMO intramuscular injection was successful and demonstrated dystrophin restoration, suggesting the possibility of DG9-PMO for DMD therapy. There was no significant toxicity observed after the injection of DG9-PMO [9].

Another study by Yokota’s research group tested the effectiveness of DG9 peptide in an SMA mouse model (Figure 4). In that study it has been shown that after a single subcutaneous administration of DG9-PMO into SMA mice (Taiwanese model), FL-SMN2 (full-length SMN2) expression was increased ~5-fold compared to unconjugated PMO in the majority of the tissues, including the brain and spinal cord. The results indicated improved motor and breathing function and muscle strength, with an increased mean survival of 58 days for DG9-PMO-treated mice, which was significantly higher compared to untreated (8 days) and unconjugated PMO-treated mice (12 days). The fact that DG9 greatly improved the uptake of PMO in the CNS and peripheral tissues at PD7, despite subcutaneous treatment at PD5, indicates that DG9-PMO can assure extensive distribution of the PMO to both the peripheral and CNS tissues. The toxicological studies show that DG9-PMO does not appear to be adverse or to impair mice’s immune systems [10].

## 9. Conclusions

Despite their huge potential, the poor biodistribution of nucleic acid-based medications has prevented them from being used clinically. As a result, cell-penetrating peptides have been thought of as a potential tactic to enhance passage across biological barriers and intracellular delivery, as covered in great detail in this chapter. However, there are a number of concerns that need to be resolved before CPPs are used in clinics, such as in vivo stability, immunogenicity, cellular toxicity, lack of selective intracellular uptake, and inability to escape from endosomes. Despite extensive research, the underlying mechanisms regulating the extravasation and cellular transport of these CPP-drug conjugates or complexes remain poorly understood. Undoubtedly, a deeper comprehension of these PPMOs’ limitations, pharmacodynamics, and mode of action will help create new CPP generations that are better able to target various tissues (and clinical diseases) and deliver their payload within the appropriate cellular compartment, giving patients hope for an improvement in their quality of life. The integration of DG9 into ASO-mediated therapy holds the potential to enhance cellular uptake and biodistribution of PMO, opening the door to more effective and precise treatments for a wide range of disorders. However, further research and development are necessary to fully realize the potential and long-term safety considerations. While DG9 has demonstrated proficiency in facilitating ASO transport into the cytoplasm, the precise underlying mechanism remains a topic requiring deeper exploration. This quest for mechanistic understanding holds promise as a compelling avenue for future investigative pursuits. With continued scientific inquiry, answers may emerge, potentially solidifying CPP-conjugated ASO-mediated therapy as a valuable asset within the arsenal of gene therapy strategies.

## Figures and Tables

**Figure 1 cells-12-02395-f001:**
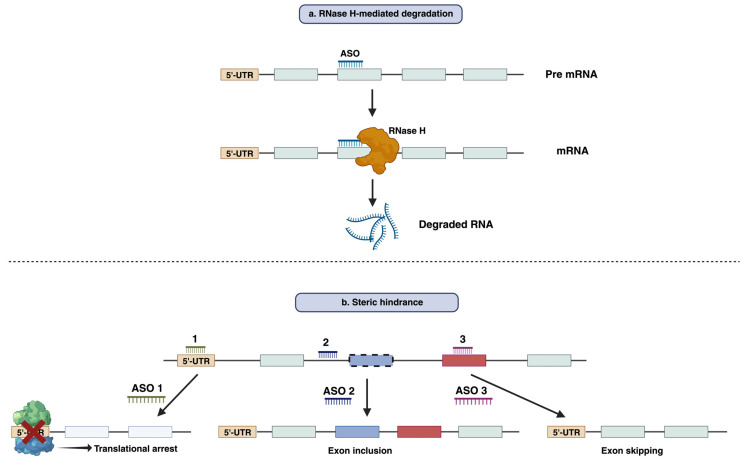
Functional mechanism of antisense oligonucleotide-mediated modulation of gene expression. (**a**) RNase H mediated degradation of RNA by antisense oligonucleotides. (**b**) Suppressing the translation or splicing modulation by an antisense oligonucleotide through steric hindrance mechanisms.

**Figure 2 cells-12-02395-f002:**
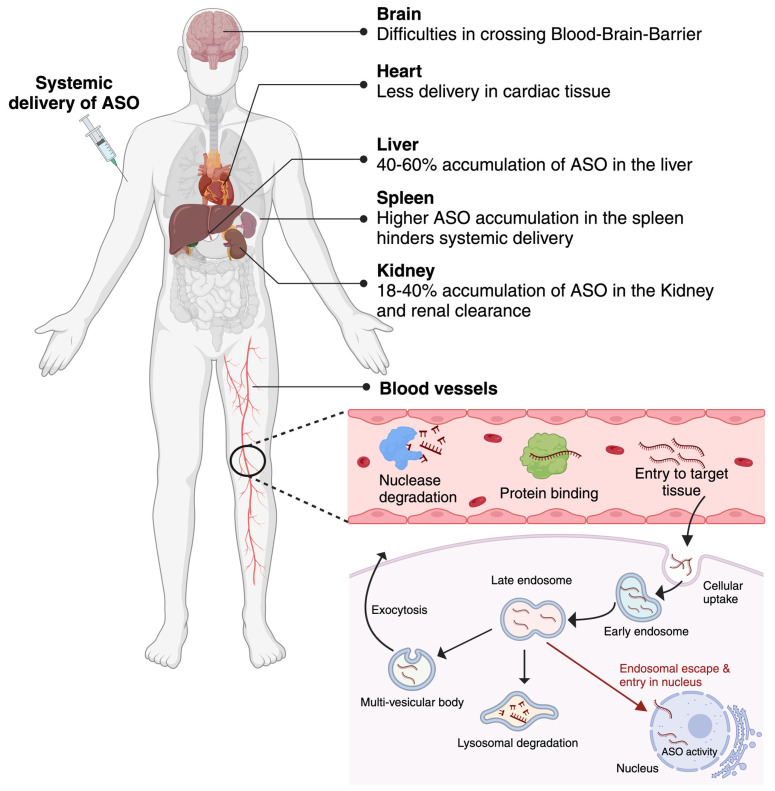
Challenges in the systemic delivery of ASO drugs. After systemic administration, antisense oligonucleotide drugs face difficulties in terms of reaching the central nervous system, heart, and other targeted tissue due to their accumulation in the liver and spleen and renal excretion. Most systemically administered, naked ASOs are susceptible to degradation by nuclease in the bloodstream. Upon reaching the target tissue, they undergo internalization by the endosome. Endosomal escape is required to exhibit activity. Apart from that, ASO may undergo degradation in the lysosome or exit the cell/tissue by the multi-vesicular system.

**Figure 3 cells-12-02395-f003:**
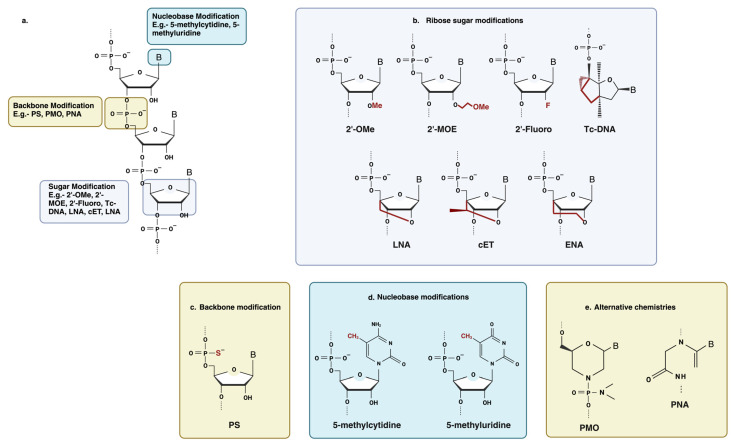
Some common chemical modifications used in antisense oligonucleotide chemistry. (**a**) Schematic of an RNA nucleotide with a common modification site. (**b**) Ribose sugar modification: 2′-OMe, 2′-O-methyl; 2′-MOE, 2′-O-methoxyethyl; 2′-Fluoro; tcDNA, tricyclo DNA; LNA, locked nucleic acid; cEt, constrained ethyl bridged nucleic acid; ENA, ethylene-bridged nucleic acid. (**c**) Backbone modification: PS, phosphorothioate. (**d**) Nucleobase modification: 5-methylcytidine, 5-methyluridine. (**e**) Alternative chemistries: PMO, phosphorodiamidate morpholino oligonucleotide; PNA, peptide nucleic acid. Created with BioRender.com (https://app.biorender.com/illustrations/64c764c0257fb4bbb5688afa (Accessed on 2 August 2023).

**Figure 4 cells-12-02395-f004:**
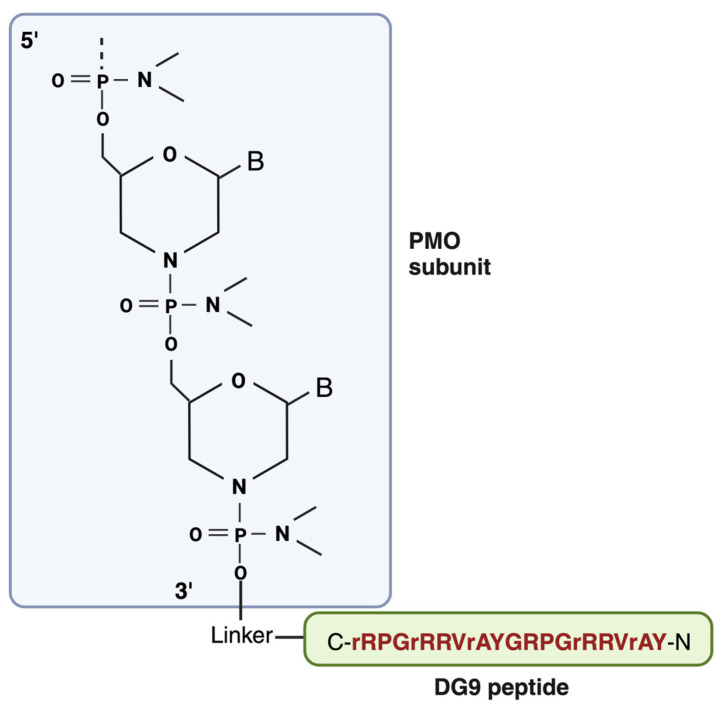
Structural representation of DG9-PMO. C-terminal end of DG9 is conjugated to the 3′end of the PMO. Created with BioRender.com (https://app.biorender.com/illustrations/64c87e5931cd4695f06c384b (accessed on 2 August 2023)).

**Table 1 cells-12-02395-t001:** FDA-approved Oligonucleotide Therapeutics. ASO, antisense oligonucleotide; dsDNA, double-stranded DNA; 2′-F, 2′-fluoro; GalNac, N-acetylgalactosamine; LNP, lipid nanoparticle; 2′-MOE, 2′-O-methoxyethyl; 2′-OMe, 2′-O-methyl; PMO, phosphorodiamidate morpholino oligonucleotide; PO, phosphodiester; PS, phosphorothioate; ROA, route of administration; siRNA, small interfering RNA; ssDNA, single-stranded DNA.

Drug Name (Market Name)	ROA	Target Gene	Indication	Modality	Chemistry	Mechanism of Action	Approval	Company
Fomivirsen (Vitravene)	Intraocular	IE-2 mRNA	Cytomegalovirus (CMV) retinitis	ASO	21 mer PS DNA	RNase H1	FDA/EMA (1998)	Ionis Pharmaceuticals (California, USA), Novartis (Basel, Switzerland)
Pegaptanib (Macugen)	Intraocular	Heparin-binding domain of VEGF-165	Neovascular age-related macular degeneration	Aptamer	27 mer 2′-F/2′-OMe pegylated	Binding and blocking	FDA (2004)	OSI Pharmaceuticals (New York, USA)
Mipomersen (Kynamro)	Subcutaneous	Apolipoprotein B100	Homozygous familial hypercholesterolemia	ASO (gapmer)	20 mer PS 2′-MOE	RNase H1	FDA (2013)	Kastle Therapeutics (Chicago, USA), Ionis Pharmaceuticals, Genzyme (Massacheusetts, USA)
Eteplirsen (Exondys 51)	Intravenous	Exon 51 of DMD	Duchenne muscular dystrophy	ASO	30 mer PMO	Splicing modulation	FDA (2016)	Sarepta Therapeutics (Massachusetts, USA)
Nusinersen (Spinraza)	Intrathecal	Exon 7 of SMN2	Spinal muscular atrophy	ASO	18 mer PS 2′-MOE	Splicing modulation	FDA/EMA (2016)	Ionis Pharmaceuticals, Biogen
Defibrotide (Defitelio)	Intravenous	Adenosine A1/A2 receptor	Veno-occlusive disease in liver	Aptamer	Mixture of PO ssDNA and dsDNA	Binding and activating	FDA (2016)	Jazz Pharmaceuticals (Ireland)
Inotersen (Tegsedi)	Subcutaneous	Transthyretin	Polyneuropathy caused by hereditary transthyretin-mediated (hATTR) amyloidosis	ASO (gapmer)	20 mer PS 2′-MOE	RNase H1	FDA (2018)	Akcea Therapeutics
Milasen *	Intrathecal	CLN7	Mila Makovec’s CLN7 gene associated with Batten disease	ASO	22 mer 2′-O-MOE, PS, 5-methyl cytosine	Splicing modulation	FDA (2018)	Boston Children’s Hospital *
Patisiran (Onpattro)	Intravenous	Transthyretin	Polyneuropathy caused by hATTR amyloidosis	siRNA (LNP formulation)	19 + 2 mer 2′-OMe modified	RNAi	FDA/EMA (2018)	Alnylam Pharma (Massachusetts, USA)
Golodirsen (Vyondys 53)	Intravenous	Exon 53 of DMD	Duchenne muscular dystrophy	ASO	25 mer PMO	Splicing modulation	FDA (2019)	Sarepta Therapeutics
Givosiran (Givlaari)	Subcutaneous	5-aminolevulinic acid synthase	Acute hepatic porphyria (AHP)	siRNA (GalNAc conjugate)	21/23 mer Dicer substrate siRNA	RNAi	FDA/EMA (2019)	Alnylam Pharma
Volanesorsen (Waylivra)	Subcutaneous	Apolipoprotein C3	Familial chylomicronemia syndrome (FCS)	ASO	20 mer PS, 2′-MOE	RNase H1	EMA (2019)	Akcea Therapeutics
Viltolarsen (Viltepso)	Intravenous	Exon 53 of DMD	Duchenne muscular dystrophy	ASO	21 mer PMO	Splicing modulation	FDA (2020)	NS Pharma
Casimersen (Amondys 45)	Intravenous	Exon 45 of DMD	Duchenne muscular dystrophy	ASO	22 mer PMO	Splicing modulation	FDA (2021)	Sarepta Therapeutics
Tofersen (Qalsody)	Intrathecal	SOD1	Amyotrophic lateral sclerosis	ASO	20 mer 2′-MOE, gapmer	RNase H1	FDA (2023)	Ionis Pharmaceuticals, Biogen
Valeriasen	Intrathecal	KCNT1	Epilepsy	ASO	2′-MOE, gapmer	RNase H1	FDA (2020)	Boston Children’s Hospital *
Atipeksen	Intrathecal	ATM	Ataxia telangiectasia	ASO		Splicing modulation		Boston Children’s Hospital *

* Milasen, Valeriasen, and Atipeksen are individualized medicines and are approved as investigational new drugs by the FDA.

**Table 2 cells-12-02395-t002:** Brief description of the most commonly used bioconjugates in the delivery of antisense oligonucleotides.

Bioconjugates	Brief Introduction	Benefits
**Lipid-based conjugates**	Lipid-based moieties are usually cholesterol and its derivatives, which are covalently conjugated to siRNA and antagomir ASOs to enhance delivery. This group of bioconjugates enhances in vivo delivery by adhering to lipoprotein particles (such as HDL and LDL) in the circulation and taking over the body’s natural system for lipid uptake and transport [101]. The overall hydrophobicity of siRNAs governs their in vivo association with the various classes of lipoprotein, with the more hydrophobic conjugates preferentially attaching to LDL and primarily being taken up by the liver. The less lipophilic conjugates preferentially bind to HDL and are consumed by the liver, adrenal glands, ovary, kidney, and small intestine. Another lipid derivative, α-tocopherol (vitamin E), was also found to increase the delivery of siRNA [12].	Improved cellular uptake. Enhanced pharmacokinetic properties. Improved cell/tissue targeting. Enhanced binding specificity. Improved in vivo stability.
**GalNac conjugates**	Trimeric GalNac is the most clinically successful tissue-targeting ligand used in ASO delivery to date. GalNAc is a carbohydrate moiety that has a high affinity for the highly expressed asialoglycoprotein receptor 1 (ASGR1 and ASPGR) [101]. This interaction promotes the endocytosis of PO ASOs and siRNAs into hepatocytes. Givosiran, a GalNAc-conjugated siRNA, was granted FDA approval for the treatment of acute hepatic porphyria in November 2019 as a result of its remarkable success [12].	
**Antibody and Aptamer conjugates**	Antibody–RNA bioconjugates offer a promising strategy for nucleic acid therapeutics; however, their utility for oligonucleotide delivery is still in the early stages of development. Antibodies are useful for the targeted delivery of oligonucleotides to cells or tissues that other methods cannot reach since they are very selective in recognizing target antigens [12,101]. Aptamers bind to their specific target proteins with high affinity, just like antibodies do. Aptamers are regarded as chemical antibodies and have demonstrated many advantages over antibodies, including being easier and less expensive to produce (i.e., through chemical synthesis), smaller size, and lower immunogenicity [12].	
**Polymer conjugates**	PEG is a non-ionic, hydrophilic polymer with a wide range of applications. It is widely used to prolong blood circulation and improve drug efficacy. PEGylation, which involves covalently adding PEG to a drug, improves the stability of ASOs and reduces renal excretion by forming a protective hydration layer around them. PEG-conjugated drugs have been found to have better pharmacokinetic and pharmacodynamic properties in terms of the drug’s chemical aspects of absorption, distribution, metabolism, excretion, and toxicity (ADMET). Other polymers besides PEG have also received attention, including poly(glycerol), poly(2-oxazoline), poly (amino acid), and poly[N-(2-hydroxypropyl)methacrylamide] because they are more ADMET-enhancing and less immunogenic [101].	
**Peptide-based conjugates**	Peptides are short chains of amino acids that can serve as carriers for oligonucleotide delivery for their cell-specific targeting, cell-penetrating, or endosomolytic properties [12]. More information about peptide conjugates is mentioned in Section 6.2.1.	

## Data Availability

Not Applicable.

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
