# Peer review of "Enhancing Antisense Oligonucleotide-Based Therapeutic Delivery with DG9, a Versatile Cell-Penetrating Peptide"

_cells, 2023, doi:10.3390/cells12192395_

Round 1

Reviewer 1 Report

“DG9, a Versatile Cell-Penetrating Peptide to Enhance Delivery 2 of Antisense Oligonucleotide-Based Therapeutics” by Umme Sabrina Haque and Toshifumi Yokota is a well organized review and deserves to be published.

Some minor points:

The line 62, “The first CPP was introduced in 1946” is misleading and not consistent with 6.2.1. Cell Penetrating Peptides and the corresponding reference to it is not correct.

P 12: “Similar to antibodies, aptamers bind to their respective target proteins with high affinity. Aptamers bind to their specific target proteins with high affinity, just like antibodies do.”  The statement is redundant, the authors should keep just one variant.

CPPs are internalized by endocytosis but also by passive mechanisms; the passive internalization of cell penetrating peptides is barely mentioned and it should be discussed.

No major issues detected.

Author Response

RESPONSE TO REVIEWERS:

We appreciate all the valuable comments from the Reviewers. We have provided additional data and have revised the manuscript according to the Reviewers’ comments and suggestions. We believe that the manuscript has been further improved. Our point-by-point response to the comments and suggestions, and the corresponding revisions and modifications in the manuscript are described below.

RESPONSE TO REVIEWER #1

“DG9, a Versatile Cell-Penetrating Peptide to Enhance Delivery 2 of Antisense Oligonucleotide-Based Therapeutics” by Umme Sabrina Haque and Toshifumi Yokota is a well organized review and deserves to be published.

Some minor points:

  1. The line 62, “The first CPP was introduced in 1946” is misleading and not consistent with 6.2.1. Cell Penetrating Peptides and the corresponding reference to it is not correct.

Response: The line has been corrected. The initial line, “The first CPP was introduced in 1946 and since then there has been a continuous effort of developing a more efficient cell-penetrating peptide that can ensure increased delivery of oligonucleotides and better pharmacological properties” is replaced with, “The initial CPP was introduced several decades ago, and ever since, there has been an ongoing endeavor to enhance cell-penetrating peptides for improved oligonucleotide delivery and enhanced pharmacological properties.”

  1. P 12: “Similar to antibodies, aptamers bind to their respective target proteins with high affinity. Aptamers bind to their specific target proteins with high affinity, just like antibodies do.” The statement is redundant, the authors should keep just one variant.

Response: The concern has been addressed and the line, “Similar to antibodies, aptamers bind to their respective target proteins with high affinity” has been removed.

  1. CPPs are internalized by endocytosis but also by passive mechanisms; the passive internalization of cell penetrating peptides is barely mentioned and it should be discussed.

Response:  We sincerely appreciate your valuable input. Your feedback on the passive internalization of cell-penetrating peptides (CPPs) is noted. We have provided a concise description of modes of CPP internalization in lines 383-388 of our manuscript, which reads: “CPPs can function either via receptor-mediated endocytosis where CPPs trigger endocytosis by directly binding to the specific cell surface receptor or via direct translocation where CPPs cross the cell by creating transient pores on the cell membrane and deliver the cargo inside the cytoplasm or nucleus”.

Reviewer 2 Report

Dr. Haque and Dr. Yokota describe cell-penetrating peptides and their ASO conjugation. This review is promising but there are some revised matters.

In the title, the name DG9 appears immediately. Shouldn't there be a full name and description in the title? Is it appropriate to include DG9 in the title because of the lengthy introduction of the FDA-approved ASO and its qualifications? I would like you to reconsider.

L. 66: R6G, PiP, and DG9 appear suddenly. The full name and its description should be mentioned because here is the first appearing place.

In Section 6, this section describes the stability and delivery of ASO. It is a bit difficult to read how DG9 is associated with this section.

Section 6.1.4: Other contents are diverse and could be more compact or subdivided?

L. 414: Only this sentence stands alone.

Section 8 is the summary of DG9. I understand that the authors want to make this point clearer, but there is much more important content except for DG9 in the review as a whole. I still think it would be better to revise the title.

Author Response

RESPONSE TO REVIEWER #2

Dr. Haque and Dr. Yokota describe cell-penetrating peptides and their ASO conjugation. This review is promising but there are some revised matters.

  1. In the title, the name DG9 appears immediately. Shouldn't there be a full name and description in the title? Is it appropriate to include DG9 in the title because of the lengthy introduction of the FDA approved ASO and its qualifications? I would like you to reconsider.

Response: We sincerely appreciate your concern. To address the query regarding the name "DG9," it is indeed a unique identifier chosen for the presence of D-arginine instead of L-arginine and due to the presence of Glycine.

Thank you for your valuable suggestion regarding the title. The primary focus of this article is elucidating how DG9 can enhance the delivery of Antisense Oligonucleotides, particularly PMO, to their target tissues. Prior to delving into the role of DG9 as a Cell-Penetrating Peptide in Antisense therapy, we deemed it essential to offer an overview of recent trends in antisense oligonucleotide-based therapy and to highlight major challenges that exist within this therapeutic approach.

We firmly believe that this contextual information is valuable for our readers as it sets the stage for appreciating the significance of DG9 as a Cell Penetrating Peptide in the field of Antisense Therapy.

  1. L. 66: R6G, PiP, and DG9 appear suddenly. The full name and its description should be mentioned because here is the first appearing place.

Response: The concern has been addressed and R6G and PiP have been removed from the keywords.

  1. In Section 6, this section describes the stability and delivery of ASO. It is a bit difficult to read how DG9 is associated with this section.

Response: Regarding your valuable suggestion about the relevance of section 6, we would like to clarify the rationale behind the structure of our article. Antisense Oligonucleotide (ASO) has undergone numerous structural modifications since it was first introduced, with the aim of enhancing its stability, specificity, cellular absorption, and toxicological profile. These modifications have indeed led to significant improvements in the pharmacokinetic properties of ASOs. However, one of the persistent challenges in the field has been achieving efficient cellular uptake, which remains a substantial barrier to the widespread use of ASOs for therapeutic purposes. In our article, we also emphasize the utilization of PMO, an alternative chemistry of ASO due to it low toxicity, in conjunction with DG9.

The central focus of our manuscript is to elucidate how DG9, as a Cell Penetrating Peptide, can enhance the cellular uptake of ASOs, particularly PMO. To provide a comprehensive foundation for this exploration, we found it pertinent to introduce our readers to the various modifications that have been undertaken to improve the pharmacokinetic properties of ASOs over time. Understanding this context helps highlight the significance of DG9 in addressing the persistent challenge of efficient cellular uptake.

  1. Section 6.1.4: Other contents are diverse and could be more compact or subdivided?

Response: The concern has been addressed and the title, “Other Oligonucleotide Modification” has been replaced by the title, “Alternative Chemistries of ASOs”.

  1. L. 414: Only this sentence stands alone.

Response: The concern has been addressed and resolved.

  1. Section 8 is the summary of DG9. I understand that the authors want to make this point clearer, but there is much more important content except for DG9 in the review as a whole. I still think it would be better to revise the title.

Response: We appreciate your concern regarding the title. We have revised the title to: Enhancing Antisense Oligonucleotide-Based Therapeutics Delivery with DG9, a Versatile Cell-Penetrating Peptide

Reviewer 3 Report

The manuscript written by Haque et al. summarizes the potential use of nucleic acids and cell penetrating peptides (CPPs) in therapy. Specifically, the authors make emphasis on phosphorodiamidate morpholino oligomers (PMOs) and DG9. The manuscript’s quality is excellent. It is well-written and organized including a balanced number of figures and tables. I have a couple of suggestions/comments to make:

Minor comments

(1) Page 2, Line 78. I think the authors have used the term “ASOs” for antisense oligonucleotides as a generic set of compounds. This might be confusing. To my mind, antisense technology is only referred to a process through which an ASO, which is made up of phosphorothioate moieties, interferes with a mRNA according to the mechanism shown in figure 1A, mainly. I think the section 2 should make mention to RNA interference as well including those ASOs and siRNAs (and other nucleic acids) that have been approved by the regulatory agencies.   

(2) Page 6, line 150. Antisense oligonucleotides have been already defined as “ASO” in the manuscript. Please, use the abbreviature.

(3)  A “future perspectives” section should be included in the manuscript.  

Author Response

RESPONSE TO REVIEWER #3

The manuscript written by Haque et al. summarizes the potential use of nucleic acids and cell penetrating peptides (CPPs) in therapy. Specifically, the authors make emphasis on phosphorodiamidate morpholino oligomers (PMOs) and DG9. The manuscript’s quality is excellent. It is well-written and organized including a balanced number of figures and tables. I have a couple of suggestions/comments to make:

Minor comments

  1. Page 2, Line 78. I think the authors have used the term “ASOs” for antisense oligonucleotides as a generic set of compounds. This might be confusing. To my mind, antisense technology is only referred to a process through which an ASO, which is made up of phosphorothioate moieties, interferes with a mRNA according to the mechanism shown in figure 1A, mainly. I think the section 2 should make mention to RNA interference as well including those ASOs and siRNAs (and other nucleic acids) that have been approved by the regulatory agencies.

Response: Additionally, in Table 1, which highlights FDA-approved oligonucleotide therapeutics, we have included a comprehensive list of these approved oligonucleotides, encompassing various categories, including siRNAs.

  1. Page 6, line 150. Antisense oligonucleotides have been already defined as “ASO” in the manuscript. Please, use the abbreviature.

Response: The concern has been addressed and “ASO” has been used instead of Antisense Oligonucleotide.

  1. A “future perspectives” section should be included in the manuscript.

Response: Thank you for your valuable suggestion regarding the inclusion of a "future perspectives" section in our manuscript. In section 9, future perspectives have been written more descriptively for better understanding. The original line, “Continued research can provide…………in the arsenal of gene therapy” has been replaced with, “While DG9 has demonstrated…………. of gene therapy strategies.”